# Nationwide introduction of HPV vaccine in Zimbabwe 2018–2019: Experiences with multiple cohort vaccination delivery

**Julie Garon Carlton** [1,2] *, **Joan Marembo** [3], **Portia Manangazira** [4], **Maxwell Rupfutse** [5], **Adelaide Shearley** [6†], **Egnes Makwabarara** [7], **Anna Hidle** [1], **Anagha Loharikar** [2]

**1** CDC Foundation, Atlanta, GA, United States of America, **2** Global Immunization Division, Centers for Disease Control and Prevention, Atlanta, GA, United States of America, **3** Expanded Program on Immunization, Zimbabwe Ministry of Health and Child Care, Harare, Zimbabwe, **4** Epidemiology and Disease Control, Zimbabwe Ministry of Health and Child Care, Harare, Zimbabwe, **5** World Health Organization Zimbabwe, Harare, Zimbabwe, **6** John Snow, Inc. Zimbabwe, Harare, Zimbabwe, **7** UNICEF Zimbabwe, Harare, Zimbabwe

† Deceased.
* jgaron@cdc.gov

**Data Availability Statement:** Data included in supplementary material. Excerpts of transcripts

## Abstract

The World Health Organization (WHO) recommends the human papillomavirus (HPV) vaccine for girls aged 9–14 years for cervical cancer prevention and encourages vaccinating multiple cohorts in the first year to maximize impact. The HPV vaccine was introduced nationwide in Zimbabwe in 2018 through a 1-week school-based campaign to multiple cohorts (all girls 10–14 years old), followed by a single cohort (grade 5 girls in school and age 10 girls out-of-school) in 2019. During the 2019 campaign, the multiple cohort's second dose was concurrently delivered with the single cohort's first dose. We interviewed national-level key informants, reviewed written materials, and observed vaccination sessions to document HPV vaccine introduction in Zimbabwe and identify best practices and challenges. Key informants included focal persons from government health and education ministries, in-country immunization partners, and HPV Vaccine Strategic Advisory Group members. We conducted a desk review of policy/strategy documents, introduction plans, readiness reports, presentations, and implementation tools. Vaccination sessions were observed in three provinces during the 2019 campaign. Key informants (n = 8) identified high cervical cancer burden, political will, vaccine availability, donor financing, and a successful pilot program as factors driving the decision to introduce the HPV vaccine nationally. The school-based delivery strategy was well accepted, with strong collaboration between health and education sectors and high community demand for vaccine identified as key contributors to this success. Challenges with transitioning from a multiple age-based to single grade- and age-based target population as well as funding shortages for operational costs were reported. Zimbabwe's first multiple cohort, school-based HPV vaccination campaign was considered successful—primarily due to strong collaboration between health and education sectors and political commitment; however, challenges vaccinating overlapping cohorts in the 2019 campaign were observed. Integration with existing health and vaccination activities

relevant to the study available upon request from Dr. Terri Hyde (thyde@cdc.gov).

**Funding:** This work was supported by Gavi, the Vaccine Alliance ["Evaluation of Human Papilloma Virus (HPV) Vaccine National Introduction in Low-and-Lower-Middle Income Countries" - Contract No. ME 9422 12 20] through a grant to CDC Foundation for project implementation. Recipient: AL. The funders had no role in study design, data collection and analysis, decision to publish, or preparation of the manuscript.

**Competing interests:** The authors have declared that no competing interests exist. Author Adelaide Shearley was unable to confirm their authorship contributions. On their behalf, the corresponding author has reported their contributions to the best of their knowledge.

and continued resource mobilization will ensure sustainability of Zimbabwe's HPV vaccination program in the future.

## Introduction

Cervical cancer is the most frequent female cancer in Zimbabwe and the leading cause of mortality reported from cancers in both sexes, with 3,043 new cases and 1,976 deaths estimated in 2020 [1]. Since 2009, the World Health Organization (WHO) has recommended human papillomavirus (HPV) vaccination for girls age 9–14 years, wherever feasible, for primary prevention of HPV infection, in addition to improvements in cervical cancer screening and treatment [2]. WHO recommends that all countries introduce HPV vaccine into the routine immunization schedule, irrespective of screening and treatment availability; countries should determine the best delivery strategies to reach optimal coverage in this target population, often not reached by routine health services. Two doses, separated by a minimum of six months, should be administered; a maximum of 12-15-month interval between doses is recommended to minimize drop-out. If feasible, countries should consider vaccinating multiple cohorts (e.g., all girls aged 9–14 years) during the first year for the greatest public health impact.

Since 2012, Gavi, the Vaccine Alliance (Gavi), has offered support for the introduction of HPV vaccine to eligible low-income countries, initially through pilot programs targeting less than 15,000 girls to understand the feasibility of HPV vaccination in the local setting and give countries an opportunity to evaluate different delivery strategies to reach a target age-group not regularly accessed through the routine immunization system [3]. Countries were intended to apply lessons learned from pilot programs to the national introduction of the HPV vaccine.

Zimbabwe conducted a successful HPV vaccine pilot program in 2014–2015 in two districts, targeting approximately 4,500 girls each year [4]. A primarily school-based campaign, selected due to high primary school enrollment, successfully reached 88% coverage in two cohorts of 10-year-old girls with a six-month interval between doses [4]. Following this positive HPV vaccine pilot program experience, Zimbabwe received Gavi support for nationwide HPV vaccine introduction in 2017.

This evaluation aimed to understand country experiences and perspectives during the first year of HPV vaccine introduction in Zimbabwe. These findings and recommendations are important for the Ministry of Health immunization program to inform program improvements for future HPV vaccine delivery. Lessons learned from Zimbabwe's national HPV vaccine introduction may help global partners gain a better understanding of the HPV vaccine national introduction, program preparation, and vaccine delivery in Gavi-eligible countries, in addition to offering useful lessons for HPV vaccine introduction in similar settings.

## Methods

Between May and September 2019, we interviewed key informants, reviewed planning documents, and observed vaccination sessions during the second vaccination campaign to document the process of HPV vaccine introduction in Zimbabwe.

### Ethics statement

The study was reviewed and determined to be non-research by the U.S. Centers for Disease Control and Prevention's Human Subject Office, and the Medical Research Council of Zimbabwe issued a program evaluation waiver. Oral informed consent was obtained by the survey

staff prior to any key informant interviews or sharing of materials and documented on the interview guide.

## Key informant interviews

We interviewed eight key informants at the national level regarding decision-making, planning, and implementation of the HPV vaccine introduction. Key informants included the immunization program manager, members of decision-making bodies and HPV vaccine focal persons from immunization partners, including WHO, United Nations International Children's Emergency Fund (UNICEF), and the U.S.-based international non-governmental organization, John Snow Inc. (JSI). A semi-structured interview guide was used to document decisions, challenges, best practices, and lessons learned (S1 File). The guide included three overarching themes of HPV vaccine introduction, including decision-making, planning and introduction, and integration and sustainability. Within these themes, questions were asked in the following areas: the drivers of decision-making for national introduction, the programmatic considerations for vaccine roll-out, financing/budgeting, coordination and Gavi application development, vaccine licensure and importation, planning, training, communication, implementation, monitoring, integration, and sustainability. One researcher conducted key informant interviews and summarized the key findings for each theme.

## Review of materials

We conducted a desk review of written materials, including policy/strategy documents, introduction plans, readiness reports, presentations, and implementation tools to gather general information related to HPV vaccine introduction in Zimbabwe. Country-developed or country-adapted written materials included program materials and evaluation reports for the pilot program, planning meeting agendas and presentations, microplanning tools, training materials, tally sheets, summary forms, registers, vaccination cards, and communications materials. Written materials were shared by the immunization program and partners before and during the key informant interviews, spanning the period before the pilot program (2012) through the 2nd national HPV vaccination campaign (2019).

## Observation of vaccination sessions

Vaccination sessions were observed during the 2019 HPV vaccination campaign (May 27–31) by three teams of external and in-country observers to further inform and document the implementation of HPV vaccination in Zimbabwe. Six districts within three (Harare, Midlands, and Masvingo) of the ten provinces in the country were purposefully selected for observing vaccination sessions based on diversity in urban/rural status and first dose HPV vaccine administrative coverage. Standardized tools adapted from the vaccination observation checklist and health worker interview guide in the WHO Post Vaccine Introduction Evaluation (PIE) Guidelines [5] were used to document observations(S1 File). The semi-structured health worker interview guide and checklist assessed vaccine administration, cold chain and safety considerations, communication and social mobilization, health worker knowledge and perceptions, and data and recordkeeping. Findings from all observations were summarized by category and reviewed along with the interview notes.

## Results

Key informant interviews at the national level included eight respondents from the immunization program, members of decision-making bodies and HPV vaccine focal persons from

immunization partners. The desk review included 17 policy/strategy documents, reports and summary presentations and over 30 country-developed tools for planning, training and implementation. A total of 19 vaccination sessions were observed over 5 days during the 2019 HPV vaccination campaign (May 27–31) by three teams of external and in-country observers (S1 Table).

## Decision-making

Respondents reported that an HPV Vaccine Strategic Advisory Group was formed to include key individuals from the Ministry of Health, Ministry of Education, WHO, UNICEF, JSI, and other partners. A National Immunization Technical Advisory Group (NITAG) was not functional at the time initial decisions around HPV vaccine introduction were made. The HPV Vaccine Strategic Advisory Group led the Gavi application process, initially through a pilot program, followed by nationwide introduction. According to the respondents, primary drivers for the national HPV vaccine introduction were high cervical cancer burden, political will, vaccine availability, availability of donor financing through Gavi, experience with a successful pilot HPV vaccine program, and high demand for HPV vaccine within the community. Strong leadership and political will for the HPV vaccine were indicated to have drove the process forward, even amid competing priorities, such as vaccine-preventable disease outbreaks. Respondents noted public concern about high rates of female cancers and evidence of high vaccine confidence throughout the country.

The Gavi application for national HPV vaccine introduction was developed by a technical working group led by the immunization program with support from the Ministry of Health, Ministry of Education, and partners. Respondents indicated that programmatic decisions were determined through a series of stakeholder meetings prior to the development of the draft introduction plan (Table 1). Vaccine supply availability was noted to have played a large role in vaccine choice from the pilot program to national introduction. Due to high school enrollment in Zimbabwe, school-based delivery was mentioned as being effective and feasible during the pilot project; therefore, the country decided to continue with school-based delivery for the national introduction. Zimbabwe implemented annual vaccination sessions with overlapping cohorts (multiple cohort's second dose was concurrently delivered with the single cohort's first dose) for national roll-out to streamline efforts and reduce costs [6]. Additional operational funds necessary for a multiple cohort introduction in the first year were provided from Gavi. Lessons learned from the pilot program were utilized in planning for the national introduction.

**Table 1. Programmatic decisions and justifications for HPV vaccine introduction in Zimbabwe.**

| Programmatic category | Decision | Justification of programmatic decision |
|---|---|---|
| Vaccine Choice | Cervarix–Bivalent HPV Vaccine, GlaxoSmithKline (GSK) | Chosen due to vaccine supply availability at the time of the application in consultation with the HPV Vaccine Strategic Advisory Group |
| Dosing Schedule | Annual dosing schedule with overlapping cohorts | Influenced by South Africa's HPV vaccine experience [7] and knowledge gained from the pilot program costing study [8] that it was less costly to vaccinate more girls at once (scale efficiency) |
| Target Age Eligibility | Multiple cohorts (girls age 10–14); single cohort (grade 5) | Viewed as an opportunity to reach as many girls as possible, Gavi support for multiple cohort vaccination was available in the first year, and the country data indicate grade 5 constituted primarily 10–year-old girls so consistency with the pilot program could be maintained |
| Delivery Strategy | School-based campaign | Selected due to high primary school enrollment rate [9], previous experience with pilot program, good support from the education sector, an existing national School Health Policy, and school health coordinators in all schools |

## Planning & implementation

Following approval of Gavi support for nationwide vaccine introduction in late 2017, national immunization program staff and partners met with provincial immunization staff, and other key stakeholders to discuss the specifics of the delivery strategy and begin planning activities. Vaccine licensure, importation, cold chain, waste management, and reporting procedures for adverse events following immunization were handled similarly to other childhood vaccines administered in Zimbabwe. Materials for HPV vaccine introduction were developed by the immunization program in the first quarter of 2018 with support from partners including JSI, WHO, and UNICEF. Materials included planning and advocacy presentations, microplanning tools, training agendas and modules, field guides, vaccination tally sheets and reporting forms, vaccination cards, and communication materials.

Microplanning at all administrative levels took place in conjunction with training and preparatory meetings throughout the country over approximately one month, according to respondents. School health coordinators (schoolteachers trained in aspects of health) developed lists of girls and maintained registers. Village health workers listed out-of-school girls and encouraged them to visit the health facility during the vaccination session or mop-up period. Health facilities maintained a list of schools in their catchment area and defined weekly activity plans. All of this information was reported to and consolidated at district, provincial and national levels to inform the national vaccine supply distribution plan. Training materials were adapted from WHO templates; cascaded training took place at the national, provincial, and district levels similar to other new vaccine introductions in Zimbabwe (training topics seen in Fig 1). Prior to the second vaccination campaign in 2019, a brief sensitization on the HPV vaccine was integrated into other routine staff meetings rather than a stand-alone training, due to budgetary constraints.

A readiness assessment, supported by international partners, took place one month before the campaign launch to determine the overall preparedness of the health system, assess community awareness or hesitancy, and identify corrective measures. This assessment found health workers to be knowledgeable about the HPV vaccination program and the handling procedures for the vaccine being used [10]. Though there were delays in the distribution of written materials, teachers, community leaders, and parents were aware of the upcoming vaccine introduction and high demand for the HPV vaccine was observed in the community.

UNICEF supported the development of a comprehensive communication plan including media briefings, advocacy meetings, radio and TV discussions, and health talks targeted at the community members visiting clinics, and the printing of social mobilization materials. Posters, key messages, and fact sheets were developed in English, Shona, and Ndebele languages. Prior to each campaign, Ministers of Health and Education participated in press briefings to inform the media and address questions. The HPV vaccine was officially launched with a public ceremony on May 2, 2018, supported by First Lady Auxillia Mnangagwa, traditional chiefs, and provincial leaders.

The first dose of the HPV vaccine was delivered to girls age 10–14 years of age through a 1-week school-based campaign (age-based) executed by teams of health workers and coordinated by the district health office. The second dose was delivered 1 year later during a similar 1-week school-based campaign concurrent with the first dose administration to a new single cohort of grade 5 girls in school and 10-year-old girls out-of-school (age- and grade-based). The HPV vaccine was available at health facilities one month after each campaign to allow any out-of-school girls or girls that may have missed a school vaccination session to receive the HPV vaccine.

Consistent with routine infant vaccinations in Zimbabwe, implied consent was used for the HPV vaccine, though some private schools continued to use written consent according to

1. Introduction to HPV infection and cervical cancer

2. HPV vaccine attributes and storage conditions

3. HPV vaccine eligibility and contraindications

4. HPV vaccine administration

5. Recording and monitoring of HPV vaccine doses

6. Communication about HPV to key stakeholders

7. Taking care of adolescent clients

**Fig 1. Training topics included during HPV vaccine introduction in Zimbabwe.**

institutional policies. Implied consent indicates that parents are provided information about upcoming vaccination activities and the physical presence of the girl at the vaccination session is considered to imply consent [11]. Immunization data were collected using vaccination cards, registers, tally sheets, and summary sheets. Supportive supervision was carried out by the Ministry of Health staff and partners before, during, and after the implementation, using a standard checklist to highlight areas of concern and identify corrective measures. The first national HPV vaccination campaign in 2018 achieved 83% coverage and administered 751,367 doses according to the WHO/UNICEF joint reporting form estimates [12, 13].

The transition from a multiple age-based cohort (girls age 10–14) to a single grade-based cohort (girls in grade 5) required a shift in both the number of cohorts (multiple to single) and

type of target group eligibility (age-based to grade-based). Observers of vaccination sessions noted confusion among health workers on eligibility during the campaign when overlapping cohorts (second dose to multi-age cohort and first dose to girls in grade 5) were targeted. Key informants interviewed reported additional challenges, including limited training in the 2019 campaign, varied information on communication materials, and the lack of clarity on recording and reporting forms. These challenges were reported to have contributed to the misunderstandings observed during vaccination sessions.

## Integration & sustainability

Zimbabwe chose to deliver the HPV vaccine through a defined 1-week campaign, and it was viewed to be resource-intensive by all eight key informants interviewed. Economic challenges in the country (i.e., inflation and multiple currencies with varying exchange rates) led to discrepancies between the budget estimations and actual costs. These challenges were intensified by fuel shortages and escalating fuel prices complicating planned transport of supplies and staff before and during vaccination campaigns leading to extension of the 2019 vaccination campaign for several days. The first year of introduction was heavily dependent on Gavi funds with funding shortfalls noted during the second campaign, contributing to implementation challenges. The sustainability of the HPV vaccination program is a key priority for the immunization program moving forward. Strategies for the continuation of the HPV program were discussed by the immunization program and partners, including combining HPV vaccination with other one-time immunization campaigns and absorbing school visits for HPV vaccine delivery into routine health facility outreach services to reduce costs.

In Zimbabwe, the Ministry of Education leadership was involved in all areas of HPV vaccine introduction, including stakeholder meetings, decision-making, Gavi application development, planning, readiness assessment, and post-introduction evaluation. School health coordinators played an important role in the HPV vaccination program by participating in training, disseminating information on HPV vaccination, identifying eligible girls, developing line lists for each school, and assisting during vaccination sessions. Stakeholders viewed school-based vaccination in Zimbabwe as being successful and credited the collaboration between health and education sectors for this success.

According to the country's HPV vaccine introduction plan, village health workers were to identify out-of-school girls in their communities, register them with school health coordinators, and encourage them to come to schools on vaccination day to receive the HPV vaccine [14]. Key informants indicated that, despite this effort, out-of-school girls were likely missed. The immunization program indicated an interest in further involving the community to identify out-of-school girls by leveraging existing programs like Reaching Every District (RED) [15] and My Village My Home [16]. These initiatives seek to establish and strengthen linkages between the community and the health system and encourage traditional leader ownership in achieving health outcomes.

In Zimbabwe, the HPV vaccination program has primarily been a stand-alone campaign during the first two years of vaccination and viewed as a resource-intensive activity. Since 2016, a School Health Policy has been in place, aiming to mainstream health topics into the school curriculum, strengthen inter-ministerial linkages/coordination and provide school-based health and nutrition services, including immunization. However, this was in the early stages of the implementation at the time of HPV vaccine introduction [17]. The immunization program is exploring the integration of the HPV vaccine into school health packages by combining vaccine delivery with other public health activities, including the delivery of other vaccines, mass drug administration, health screening, and hygiene education.

## Discussion

Though a range of HPV vaccine delivery strategies can be successful in low-resource settings, schools have been shown to be highly effective at reaching large numbers of adolescent girls and achieving high coverage, with the engagement of school personnel an essential component to success [18–21]. The introduction of the HPV vaccine in Zimbabwe illustrates a successful collaboration between the government health and education sectors, resulting in effective school-based vaccine delivery and high coverage. In addition to good collaboration between partners, strong political commitment is essential for a successful HPV vaccination program [19]. Inter-ministerial support was seen throughout the planning and introduction process, and the First Lady and community leaders supported a highly publicized launching ceremony.

Several operational challenges were observed with the change in target eligibility criteria during the 2019 HPV vaccination campaign, illustrating the importance of clear messaging, detailed guidelines, and straightforward reporting procedures within country programs. There are advantages and disadvantages to both age- and grade-based eligibility criteria; the choice is largely dependent upon country context. Experience from HPV vaccination programs in low- and middle-income countries has shown grade-based eligibility can be more feasible to implement than age-based eligibility when delivering this vaccine in schools; however, when data systems were organized around age-based data, estimating coverage can be more difficult [19, 22, 23]. Differing eligibility criteria for in-school and out-of-school target groups adds complexity to communication programs. Countries planning for the HPV vaccine introduction should consider the challenges associated with eligibility criteria and programmatic transitions during the early stages of the introduction and evaluate the capacity of communication, training, and recording/reporting systems to adapt to these changes in a newly established program.

Zimbabwe estimates a very high primary school enrollment (up to grade 7) at 97.5% while secondary school enrollment is estimated at only 53%, according to the Ministry of Education data sources [9]. While school-based strategies can be successful at reaching large amounts of girls at one time, the difficulty of enumerating and vaccinating out-of-school girls has been noted in HPV vaccination programs in low- and middle-income countries [19]—a concern echoed by the stakeholders in this evaluation. Alternative strategies to school-based platforms include health-facility vaccination and community outreach. Future efforts also include developing specialized plans to reach girls living with HIV with an additional dose of the HPV vaccine, as three doses are recommended for this population by WHO [24, 25]. Country immunization staff should continue exploring community linkages to identify out-of-school girls and consider ways to scale up efforts to reach other special populations in future vaccination campaigns.

Though school-based campaigns are capable of achieving high coverage in a short period, they have been documented to be resource-intensive [18]. Funding shortages in the 2019 campaign led to reductions in activities related to training, communication, and preparation which contributed to challenges in determining eligibility and recording vaccinations administered. To ensure the maintenance of gains achieved so far, resource mobilization for future campaigns in Zimbabwe is vital. During pilot programs, countries reported considerable uncertainty about financing availability in the future, due to high HPV vaccine costs and high perceived delivery costs associated with school-based delivery [18, 19]. Future market entry of additional HPV vaccine products may drive down vaccine costs and a potential single-dose HPV vaccine recommendation by WHO in the coming years would further reduce the financial burden for low- and middle-income countries to introduce and sustain HPV vaccination [24–27]. In the meantime, integration with other vaccination or school health activities and

other cost-saving measures may reduce resource requirements for HPV vaccine delivery in Zimbabwe and elsewhere, but it remains unknown if this would have programmatic implications.

Given the unique characteristics of Zimbabwe and the limited number of key informants interviewed, the generalizability of the findings may be limited, though general concepts may be useful for countries planning HPV vaccine introduction. Only one researcher was responsible for reviewing documents and conducting interviews; therefore, comparability and confirmation of results was not done. Lastly, interviews took place a year after the HPV vaccine introduction, and there might have been some poor recall or staff turnover.

## Conclusion

Good collaboration between the health and education sectors, detailed preparation, and strong political commitment led to the successful introduction of the HPV vaccine through a school-based campaign in Zimbabwe. Lessons learned in Zimbabwe, including challenges associated with changing eligibility criteria, may be considered by other countries planning to introduce the HPV vaccine, nationally. Exploring ways to integrate the HPV vaccine with other vaccination or school health activities and prioritizing sustained funding for HPV vaccination in Zimbabwe will contribute to a generation with reduced risk of cervical cancer and the associated deaths. Zimbabwe's success in delivering the HPV vaccine to a multiple cohort of girls shows what can be achieved in the face of many challenges and can serve as an example to other countries embarking on similar endeavors.

## Supporting information

**S1 File. Zimbabwe national-level interview guide and health worker session observation questionnaire.**
(PDF)

**S1 Table. Summary table of health facility-level hpv post introduction evaluation findings.**
(XLS)

## Acknowledgments

Immunization program manager and staff at Zimbabwe Ministry of Health and Childcare, the Ministry of Primary and Secondary Education, City of Harare, WHO, UNICEF, John Snow International and Médecins Sans Frontiéres, participants of the Post Introduction Evaluation, including representatives from the CDC, WHO, and Gavi.

## Author Contributions

**Conceptualization:** Julie Garon Carlton, Portia Manangazira, Maxwell Rupfutse, Anna Hidle, Anagha Loharikar.

**Data curation:** Julie Garon Carlton.

**Formal analysis:** Julie Garon Carlton.

**Funding acquisition:** Anagha Loharikar.

**Investigation:** Julie Garon Carlton, Joan Marembo, Portia Manangazira, Maxwell Rupfutse, Adelaide Shearley, Egnes Makwabarara, Anna Hidle.

**Methodology:** Julie Garon Carlton, Joan Marembo, Maxwell Rupfutse, Anagha Loharikar.

**Project administration:** Julie Garon Carlton, Maxwell Rupfutse, Anna Hidle.

**Supervision:** Joan Marembo, Portia Manangazira, Anagha Loharikar.

**Validation:** Adelaide Shearley, Egnes Makwabarara.

**Writing – original draft:** Julie Garon Carlton.

**Writing – review & editing:** Anna Hidle, Anagha Loharikar.

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
