## [Decision Letter · Decision Letter 0]

13 Oct 2021

PGPH-D-21-00325

Nationwide introduction of HPV vaccine in Zimbabwe 2018-2019: Experiences with multiple cohort vaccination delivery

Dear Dr. Carlton,

Thank you for submitting your manuscript to PLOS Global Public Health. After careful consideration, we feel that it has merit but does not fully meet PLOS Global Public Health’s publication criteria as it currently stands. Therefore, we invite you to submit a revised version of the manuscript that addresses the points raised during the review process.

We look forward to receiving your revised manuscript.

Kind regards,

Anat Rosenthal

Academic Editor

Journal Requirements:

1. Please include a copy of the interview guide used in the study, in both the original language and English, as Supporting Information, or include a citation if it has been published previously.

2. We note that participants provided oral consent. Please state in the Methods:

- Why written consent could not be obtained

- How oral consent was documented

For more information, please see our guidelines for human subjects research: https://journals.plos.org/plosone/s/submission-guidelines#loc-human-subjects-research

3. We noticed that you used “data not shown”/"unpublished data" in the manuscript. We do not allow these references, as the PLOS data access policy requires that all data be either published with the manuscript or made available in a publicly accessible database. Please either remove these references, or amend the supplementary material to include the referenced data.

4. We do not publish any copyright or trademark symbols that usually accompany proprietary names, eg (R), (C), or TM  (e.g. next to drug or reagent names). Therefore please remove all instances of trademark/copyright symbols throughout the text, including CervarixTM on table 1.

5. In the online submission form, you indicated that "Excerpts of transcripts relevant to the study available upon request". All PLOS journals now require all data underlying the findings described in their manuscript to be freely available to other researchers, either 1. In a public repository, 2. Within the manuscript itself, or 3. Uploaded as supplementary information.

6. Please amend your detailed Financial Disclosure statement. This is published with the article, therefore should be completed in full sentences and contain the exact wording you wish to be published.

i). State the initials, alongside each funding source, of each author to receive each grant.

ii). State what role the funders took in the study. If the funders had no role in your study, please state: “The funders had no role in study design, data collection and analysis, decision to publish, or preparation of the manuscript.”

Additional Editor Comments (if provided):

The article is informative, interesting and scientifically sound and the authors' analysis has the potential to benefit similar programs in other countries. Some minor revisions are required in order to prepare this article for publication following reviewers' comments.

Reviewers' comments:

Reviewer's Responses to Questions

**Comments to the Author**

1. Does this manuscript meet PLOS Global Public Health’s publication criteria? Is the manuscript technically sound, and do the data support the conclusions? The manuscript must describe methodologically and ethically rigorous research with conclusions that are appropriately drawn based on the data presented.

Reviewer #1: Yes

Reviewer #2: Yes

2. Has the statistical analysis been performed appropriately and rigorously?

Reviewer #1: N/A

Reviewer #2: N/A

3. Have the authors made all data underlying the findings in their manuscript fully available (please refer to the Data Availability Statement at the start of the manuscript PDF file)?

Reviewer #1: Yes

Reviewer #2: No

4. Is the manuscript presented in an intelligible fashion and written in standard English?

Reviewer #1: Yes

Reviewer #2: Yes

5. Review Comments to the Author

Reviewer #1: 

The authors present an analysis of Zimbabwe’s nationwide introduction of the HPV vaccine through key informant interviews, written material review, and observation of vaccination sessions. The manuscript is informative, and presents useful lessons learned, as well as various perspectives of which strategies contributed to the success of the vaccination strategy in Zimbabwe. In particular, table 1, as well as the integration and sustainability sections are quite informative. The paper is overall well-written, the study design is scientifically sound despite a limited number of key informant interviews, and the results are presented in a concise manner. There are minor grammatical errors and tense shifting that makes parts of the paper difficult to read, and could be improved for better understanding. A few examples are given below.

Minor comments:

Below are some other minor revisions which I think could help improve the manuscript: 

Abstract: 

 Results: Line 48-49 should be either broken up or explained more succinctly. The age-based vs. Grade-based is it's own topic (strategy). Funding shortages and sustainability are future challenges to the current approach, as then discussed in the conclusions paragraph?

Introduction:

 Line 75: »high school enrollment« is confusing for those from places high school is the term used for secondary school. Rephrasing will help clarify the ages/grades targeted for vaccinations.

Results:

This may not be available, but would improve the manuscript: what percentage of eligible girls were successfully vaccinated and how many were missed?

Discussion:

Lines 291-294: What are strategies that are less resource intensive that could be considered as an alternative to the school-based strategies?

Line 304: While Zimbabwe is unique, why can’t some of these concepts be generalized amd used elsewhere for vaccine implementation strategy?

Reviewer #2: This is a very interesting and well-written paper. I enjoyed reading it. We need to share these experiences. The success of the initiative in Zimbabwe is probably inspired by the preparations and supportive steps taken and described. This is in contrast to the problems being encountered in many places with the roll out of vaccines for Covid-19. I have no required changes, just a pet thing about data "are" rather than data "is". But I realise I am in a minority.

6. PLOS authors have the option to publish the peer review history of their article (what does this mean?). If published, this will include your full peer review and any attached files.

**Do you want your identity to be public for this peer review?** For information about this choice, including consent withdrawal, please see our Privacy Policy.

Reviewer #1: No

Reviewer #2: No

---

## [Decision Letter · Decision Letter 1]

23 Feb 2022

Nationwide introduction of HPV vaccine in Zimbabwe 2018-2019: Experiences with multiple cohort vaccination delivery

PGPH-D-21-00325R1

Dear Mrs. Carlton,

We are pleased to inform you that your manuscript 'Nationwide introduction of HPV vaccine in Zimbabwe 2018-2019: Experiences with multiple cohort vaccination delivery' has been provisionally accepted for publication in PLOS Global Public Health.

Best regards,

Anat Rosenthal

Academic Editor

The manuscript is improved and all the comments from the reviewers were addressed. Both reviewers have minor comments to be considered in the final version of the article (such as making the data available, and addressing the stigma and the potential future developments in vaccine delivery). There are also a few grammatical errors to be addressed in the final version. Congratulations on a very interesting article.

Reviewer Comments (if any, and for reference):

Reviewer's Responses to Questions

**Comments to the Author**

1. If the authors have adequately addressed your comments raised in a previous round of review and you feel that this manuscript is now acceptable for publication, you may indicate that here to bypass the “Comments to the Author” section, enter your conflict of interest statement in the “Confidential to Editor” section, and submit your "Accept" recommendation.

Reviewer #1: All comments have been addressed

Reviewer #2: All comments have been addressed

2. Does this manuscript meet PLOS Global Public Health’s publication criteria? Is the manuscript technically sound, and do the data support the conclusions? The manuscript must describe methodologically and ethically rigorous research with conclusions that are appropriately drawn based on the data presented.

Reviewer #1: Yes

Reviewer #2: Yes

3. Has the statistical analysis been performed appropriately and rigorously?

Reviewer #1: Yes

Reviewer #2: N/A

4. Have the authors made all data underlying the findings in their manuscript fully available (please refer to the Data Availability Statement at the start of the manuscript PDF file)?

Reviewer #1: Yes

Reviewer #2: Yes

5. Is the manuscript presented in an intelligible fashion and written in standard English?

Reviewer #1: Yes

Reviewer #2: Yes

6. Review Comments to the Author

Reviewer #1: The manuscript is improved. There are a number of minor gramattical errors that should be addressed prior to publication. Furthermore, there is some mixing of discussion in the results, which could be tightened up a bit more to ensure the results contain just that. The recommendations are very useful, and certainly a highlight of the paper. One thing not mentioned (potentially not addressed here) is that there is not as much of the stigma attached to the vaccine as in other places. The other potential recommendation is the hope that one day a vaccine that doesn't require cold chain and could be incorporated into other programs could be a huge benefit in these settings.

Reviewer #2: Thank you I am looking forward to the article being published.

I have presumed that the authors have made their data accessible and that there will be an appropriate link in the published article.

7. PLOS authors have the option to publish the peer review history of their article (what does this mean?). If published, this will include your full peer review and any attached files.

**Do you want your identity to be public for this peer review?** For information about this choice, including consent withdrawal, please see our Privacy Policy.

Reviewer #1: No

Reviewer #2: No
